# The Evolution of tRNA Copy Number and Repertoire in Cellular Life

**DOI:** 10.3390/genes14010027

**Published:** 2022-12-22

**Authors:** Fenícia Brito Santos, Luiz-Eduardo Del-Bem

**Affiliations:** 1Del-Bem Lab, Department of Botany, Institute of Biological Sciences (ICB), Federal University of Minas Gerais (UFMG), Belo Horizonte 31270-901, MG, Brazil; 2Graduate Program in Bioinformatics, Institute of Biological Sciences (ICB), Federal University of Minas Gerais (UFMG), Belo Horizonte 31270-901, MG, Brazil

**Keywords:** tRNA, CNV, genome evolution, eukaryogenesis

## Abstract

tRNAs are universal decoders that bridge the gap between transcriptome and proteome. They can also be processed into small RNA fragments with regulatory functions. In this work, we show that tRNA copy number is largely controlled by genome size in all cellular organisms, in contrast to what is observed for protein-coding genes that stop expanding between ~20,000 and ~35,000 *loci* per haploid genome in eukaryotes, regardless of genome size. Our analyses indicate that after the bacteria/archaea split, the tRNA gene pool experienced the evolution of increased anticodon diversity in the archaeal lineage, along with a tRNA gene size increase and mature tRNA size decrease. The evolution and diversification of eukaryotes from archaeal ancestors involved further expansion of the tRNA anticodon repertoire, additional increase in tRNA gene size and decrease in mature tRNA length, along with an explosion of the tRNA gene copy number that emerged coupled with accelerated genome size expansion. Our findings support the notion that macroscopic eukaryotes with a high diversity of cell types, such as land plants and vertebrates, independently evolved a high diversity of tRNA anticodons along with high gene redundancy caused by the expansion of the tRNA copy number. The results presented here suggest that the evolution of tRNA genes played important roles in the early split between bacteria and archaea, and in eukaryogenesis and the later emergence of complex eukaryotes, with potential implications in protein translation and gene regulation through tRNA-derived RNA fragments.

## 1. Introduction

Transfer RNA (tRNA) are essential molecules for protein synthesis, performing a key role in the gene decoding process in all cellular life. tRNA genes are present in multigene families comprising near-identical copies spread throughout the genome as multiple *loci* [1]. These *loci* are polymerase III (Pol-III) transcribed into structured non-coding RNAs that carry specific amino acids into the catalytic site of ribosomes, allowing proteins to be synthesized from mRNA information [2,3]. Therefore, every cell needs a minimal pool of tRNA molecules for protein synthesis. The evidence suggests that tRNA are among the earliest biomolecules to ever emerge in biological systems, as they are essential in the flow of information from genes to proteins in every cell [4,5,6,7,8,9,10,11,12,13].

Algorithms developed for the prediction of tRNA genes in genomes allowed for the annotation of a vast number of candidate *loci* for a multitude of species belonging to all domains of cellular life [14,15]. The total copy number (CN) of tRNA *loci* varies widely among species [1,2,16,17,18,19,20]. Even related species, which otherwise show small differences in genomic features such as genome size, number and length of protein-coding genes, and shared intron positions [1,16,21], can present substantial tRNA copy number variation (CNV), as seen in primates, where great apes have ~600 tRNAs and some galagos (*Galagidae*) have tens of thousands of tRNAs genes [17].

CNV is a form of structural variation in which the number of copies of a particular chromosome segment varies between individuals, populations or species [22]. This variation includes loss and amplification of particular genes, and some examples have been reported for a vast number of protein-coding and non-coding genes in all domains of life [1,16,23,24,25,26]. CNVs may lead to changes in gene expression that can be positively selected in populations [26], fixing the newly duplicated copies. CNVs are important for rapid adaptative responses to stress and are also linked to disease [24,26,27,28,29,30,31].

Intraspecific variation in the tRNAome (pool of tRNA genes in a genome) is known among strains of yeast, where they seem to have evolved concerted with codon usage bias on the transcriptome [18,21,32,33]. CNVs that produce an uneven supply of some tRNA drive a selective pressure in the transcriptome possibly leading to adaptive evolution of the codon usage [18].

Although tRNA CNV is known to exist between species [1,2,16,18,20,21], less is known about macroevolutionary patterns that might shape the repertoire of tRNA gene families across the three domains of cellular life. In this work, we analyzed the tRNA repertoire of nearly 5000 genomes to investigate the underlying patterns of diversity and abundance of tRNA genes in cellular organisms. Our results indicate that the CN of tRNA genes expands as a function of genome size, in a pattern different from what is observed for protein-coding genes. We discuss how the increase in tRNA CN and the anticodon repertoire was likely an important factor during eukaryogenesis and how intraspecific tRNA CNV might affect genome function at an individual level.

## 2. Materials and Methods

### 2.1. tRNA Data Collection

tRNA full annotation data, along with genomic and mature sequences, were downloaded from the Genomic tRNA Database (http://gtrnadb.ucsc.edu/) [15], accessed on 22 June 2022. Annotation data included the tRNA anticodon, isotype, number of introns, gene locus coordinates and other features identified by tRNAscan-SE [14]. Taxonomic information associated with each tRNA was retrieved from the NCBI Taxonomy using Taxallnomy [34]. Genomes with fewer than 20 annotated tRNA isoacceptors were removed from the analysis (25 out of 4803 or ~0.52% of the sample).

### 2.2. tRF Data Collection

tRF data were downloaded from the tRFdb (http://genome.bioch.virginia.edu/trfdb/) on 2 September 2022 [35].

### 2.3. Statistical Analyses and Data Manipulation

All statistical analyses and data manipulation were conducted with R version 4.1.0, using in-house scripts. Welch’s two-sample *t*-test and the Wilcox Mann–Whitney U test were used to access the statistical significance when comparing groups of data. The first was used to test whether the means were different, and the second was applied to test whether the medians were different. Differences were considered significant for *p*-values < 0.05. All plots were generated with ggplot2 (v.3.3.5) [36] and pheatmap (v.1.0.12) [37] packages.

## 3. Results

### 3.1. tRNA Copy Number Variation in Cellular Organisms

Besides being found in all cells, tRNA genes vary extensively in copy number between species. To further investigate this, we decided to quantify the number of tRNA *loci* in 4803 completely sequenced genomes [15]. The vast majority of these were bacteria species (4027 or ~84% of the whole sample). We also used 217 archaean genomes (~4.5%) and 559 from eukaryotes (~11%). Among the 489,672 predicted tRNA *loci* across all genomes, we noticed a substantial difference in terms of mean tRNA gene count per genome between eukaryotes and prokaryotes. Eukaryotes have a mean of 403 ± 27.1 SE tRNA *loci* per genome (Figure 1A) with extreme cases varying up to three orders of magnitude between species (median = 200.00, IQR = 279.00) as seen for the jelly fungus *Tremella mesenterica* (40 tRNA loci) and the bony fish *Danio rerio* (10,471 tRNA *loci*) (Figure 1B). Our data suggest that bacteria have significantly more tRNA genes (mean = 60.30 ± 0.315 SE) when compared to archaea (mean = 47.60 ± 0.367 SE), which were ~6.0 and ~7.5 times less, respectively, than in an average eukaryote (Figure 1A). tRNA gene count per genome shows a variation of one and a half orders of magnitude between bacterial species (median = 57.00, IQR = 28.00), while archaeans present only a narrow variation (median = 47.00, IQR = 3.00) (Figure 1B).

tRNAs can be further classified based on the amino acid they carry, giving rise to at least 20 different tRNA isoacceptor families, one for each essential amino acid, or based on the anticodon they have (64 tRNA possibilities). The anticodons used to encode each amino acid vary between species, giving rise to specific combinations of anticodon and amino acids that will be further explored later. The total repertoire of tRNA anticodons found in each genome differs among domains. Bacteria had the lowest diversity of tRNA anticodons per genome (mean = 37.20 ± 5.03 SD; median = 39.00, IQR = 8.00). Archaea showed an intermediary mean of 42.40 ± 3.59 SD (median = 44.00, IQR = 1.00), while Eukarya showed the greatest diversity of tRNA anticodons per genome (mean = 47.3 ± 6.97 SD; median = 45.00, IQR = 6.00) (Figure 1C). Bacteria and eukaryotes presented a great variation in the anticodon repertoire, while archaean genomes varied only slightly (Figure 1C). To investigate this variation, we further divided the data according to taxonomic groups within each domain (Figure 1D), showing that genomes from bacterial phyla such as *Fusobacteria* and *Tenericutes* usually have less than 30 different tRNA anticodons available for protein synthesis, while genomes from the phylum Chloroflexi usually possess ~45 different tRNA anticodons. Surprisingly, archaean genomes from different phyla, such as *Euryarchaeota* and *Thaumarchaeota*, have a diversity of tRNA anticodons comparable to what is observed in Fungi and ‘Protozoa’ (Figure 1D). Animals and land plants, on the other hand, are the organisms with the greatest diversity of tRNA anticodons, usually between 50 and 60. Invertebrate genomes usually have a lower diversity of tRNA anticodons (mean = 52.60 ± 6.99 SD; median = 49.00, IQR = 13.00) when compared to vertebrates (mean = 60.50 ± 3.63 SD; median = 62.00, IQR = 5.00), with land plants appearing somewhere in between (mean = 56.50 ± 4.56 SD; median = 57.00, IQR = 6.00) (Figure 1D). Besides the theoretical maximum of 64 anticodons, our data indicate that most genomes have consistently fewer than 50 different tRNA anticodons, suggesting that only in particular contexts, such as those seen for complex eukaryotes (e.g., land plants and vertebrates), a vaster tRNA anticodon repertoire evolves and is maintained.

We also calculated the mean tRNA gene count for each amino acid and anticodon for eukaryotes, bacteria, and archaea. We noticed that eukaryotic genomes present a significantly higher number of tRNA genes per amino acid and anticodon compared to prokaryotes (Figure 1E,F). In our sample, eukaryotic genomes had a median of 17.6 tRNA genes per amino acid (including the rare Selenocysteine) and 7.12 per anticodon. By contrast, the median number of tRNA genes per amino acid was 2.90 for bacteria and 2.10 for archaea. The median per anticodon was 1.40 for bacteria and 1.11 for archaea (Figure 1E,F). Our results indicate that eukaryotes have a significantly higher number of tRNA genes dedicated to each amino acid and/or anticodon compared to prokaryotes, which could indicate that the process of eukaryogenesis involved a substantial expansion of tRNA gene families.

### 3.2. Universalities and Particularities in the Evolution of Translational Decoding by tRNAs

Since our data is based on tRNA gene prediction by tRNAscan-SE across complete genomes, herein we use the DNA notation when referring to the tRNA anticodons. In our sample, we found 63 anticodons that were assigned to at least one of the 21 proteinogenic amino acids. The only anticodon that was never associated with any amino acid in any organism was TTA. However, ^Sup^tRNA_(TTA)_ is used by a few eukaryotes (21.65%) and rare bacteria (0.79%) as a suppressor tRNA [38,39]. Besides a core of conserved tRNAs, some unusual combinations of anticodons and transported amino acids are found in specific taxa. For each domain, we found a different number of observed tRNA anticodon–amino acid combinations. In the eukaryote genome set, we found 69 different combinations, while in bacteria 71. The archaean genomes had the lowest number of combinations (51), indicating a more conserved association between tRNA anticodons and the amino acid they carry (Figure 2A).

Despite the diversity of amino acid–anticodon combinations observed, some anticodons are preferentially used over others to transport each amino acid. Figure 2B,C shows the frequency of genomes that display each type of tRNA per domain and the mean copy number of each tRNA gene per domain, respectively. We further classified the tRNAs into groups using Greek letters, depending on the degree of universality of each amino acid–anticodon combination. The α-group consists of 26 combinations present in more than 70% of the genomes in all domains. These tRNAs are sufficient to transfer 19 out of the 20 standard amino acids. The only exception is the ^Ile^tRNAs. The ^Ile^tRNA_(CAT)_ is exclusively found in prokaryotes as an alternative start anticodon [40]. Besides that, the ^Ile^tRNAs can have three other anticodons, with ^Ile^tRNA_(GAT)_ being universally found in prokaryotes (ARC = 100%, BAC = 97.96%) and uncommon in eukaryotes (EUK = 31.48%), whilst ^Ile^tRNA_(AAT)_ and ^Ile^tRNA_(TAT)_ are nearly universal in eukaryotic genomes (97.85% and 95.70%, respectively) but rare in prokaryotes (ARC = 0.92%, BAC = 1.41%). Other amino acids show the same pattern, indicating that some anticodons are preferentially used while others are avoided in each domain. The β-group integrates tRNAs with combinations of amino acids and anticodons that are very frequent in eukaryotes and archaea and are mostly avoided by bacteria (Figure 2B,C). The γ-group is composed of amino acid–anticodon combinations that are frequently found in prokaryote genomes and are mostly avoided in eukaryotes, while the δ-group contains tRNAs preferably found in eukaryotes and avoided by bacteria and archaea (Figure 2B,C). The ε-group contains tRNAs that are rare or absent in all contexts, some of which are ^Sup^tRNAs. The ^Arg^tRNA_(ACG)_ is the only case shared by most eukaryotic and bacterial genomes, which is rare in archaeal genomes. Archaea frequently have the ^Arg^tRNA_(GCG)_, which is rare in eukarya and bacteria (Figure 2B,C). Surprisingly, the ^Met^tRNA_(CAT)_ is the most abundant tRNA gene in all domains, with a mean of ~17.60 copies per eukaryotic genome, ~2.21 per archaeal genome and ~3.44 copies per bacterial genome (Figure 2C).

Figure 2D shows that the tRNA genes that are the most frequent in each genome are usually the most pervasive across species in each domain, indicating that the tRNA genes with the most universal combinations of anticodons and transported amino acids are preferably amplified in CN in all cellular lineages. If we take the tRNAs that are shared by at least 70% of genomes in each domain, around half of them (26 tRNAs) are shared by all cells (Figure 2E). All common tRNAs in bacteria are also common in archaea and/or in eukaryotes. Two tRNAs are mostly found in archaea and eight in eukarya (Figure 2E). The overlap between eukaryotes and bacteria is restricted to one common tRNA, while eukaryotes have 10 common tRNAs shared with archaea (Figure 2E).

### 3.3. tRNA Gene Density per Genome Shows an Overlap between Single-Celled Eukaryotes and Prokaryotes

If we consider the absolute number of protein-coding genes, eukaryotes tend to have up to one order of magnitude more genes than prokaryotes. However, when we analyze the gene density per Mb, eukaryotes tend to have genomes rich in non-coding DNA, resulting in small gene density, while prokaryotes have gene-rich genomes. In fact, eukaryotic genomes rarely have more than 500 protein-coding genes per Mb (Figure 2F), while bacteria and archaea have densities around 1000 genes per Mb, with archaeal genomes being usually a little more gene dense (Figure 2F). Almost no overlap is observed between eukaryote and prokaryote gene density per genome (Figure 2F). On the other hand, tRNA gene density is rarely over 40 *loci* per Mb and shows a clear overlap between eukaryotes and prokaryotes (Figure 2G). While tRNA gene density distribution is very similar between archaea and bacteria, eukaryotes tend to have <5 tRNA genes per Mb while bacteria and archaea typically have ~20 tRNA genes per Mb (Figure 2G). To further reveal taxonomic bias, we split eukaryotes according to the main lineages, revealing that vertebrate genomes tend to have very low tRNA density per Mb, followed by land plants and invertebrates. Interestingly, fungi exhibit a high degree of tRNA density variation, with a significant proportion of genomes with having >10 tRNAs/Mb (Figure 2H). Our results indicate that part of fungal genome diversity has a density of tRNA genes comparable to what is observed in prokaryotes, while vertebrates and land plants have large genomes with exceedingly few tRNAs per Mb (Figure 2G,H).

### 3.4. tRNA CNV and Diversity Is Shaped by Genome Size Expansion

We next wondered whether the CN and diversity of tRNA genes of a given genome is correlated with other genomic parameters such as genome size and number of protein-coding genes. We observed that genomes with a higher number of protein-coding genes (usually >2 × 10^4^ genes) have substantially more tRNA genes when compared with genomes with fewer genes (10^3^–10^4^ genes) (Figure 3A). The CN of tRNA genes is well correlated with genome size, following a pattern of distribution quite different from that of protein-coding genes. The number of protein-coding genes expands fast when genome size increases from 1 to 10 Mb, a decrease in expansion is observed for genomes between 10 and 100 Mb and little if any expansion is observed for genomes >>100 Mb, where the haploid set of protein-coding genes is usually under 3.5 × 10^4^ genes. By contrast, the number of tRNA genes continues to expand at the same pace in genomes >>100 Mb, commonly reaching 10^3^ to 10^4^ copies in genomes >1 Gb (Figure 3B). For eukaryotes, our data suggest that tRNA CN expands faster than protein-coding genes when the genome increases in size, mostly due to whole genome duplications (WGD) (Figure 3C). This pattern is different from what we observed in prokaryotes, where genome expansion seems to be coupled with an increase in the number of protein-coding genes preferably over tRNAs (Figure 3D,E).

Considering the repertoire of tRNAs per genome, we observed a positive relationship between the number of different isoacceptors (anticodon repertoire) and the number of tRNAs in each genome (Figure 3F). Organisms with a small tRNA gene count such as bacteria tend to have fewer anticodons available in the tRNA pool. Archaea are a particular case, for they usually have fewer tRNA genes than bacteria and have a significantly greater anticodon diversity, indicating lower tRNA gene redundancy (Figure 1D and Figure 3F). On the other extreme, land plants and vertebrates tend to have significantly more tRNA gene copies and are the organisms that sustain the largest diversity of tRNA anticodons per genome (Figure 1D and Figure 3F). We also observed genome size as an interesting predictor of the anticodon diversity of a genome. Organisms with larger genomes tend to have a more diverse tRNA pool (Figure 3G). The total number of protein-coding genes found in a genome has an interesting relationship with the diversity of the anticodon. Organisms that cross a threshold of around 14,000 genes seem to experience an accelerated increase in anticodon diversity, as seen mostly for vertebrates and land plants, which consistently tend to have a repertoire of >50 anticodons (Figure 3H).

In our study, we also wondered about the accumulation of tRNA pseudogenes. These are defined as genomic *loci* that can be identified by tRNAScan-SE without the assignment of an anticodon; they most likely are unable to produce functional mature tRNA. The number of such elements per genome follows a pattern of accumulation that is nearly identical to what is observed for possibly functional tRNA genes, except that they are usually 1 to 2 orders of magnitude less abundant per genome (Figure 3I). We also investigated gene and mature tRNA length per domain (Figure 3J). Our data suggest that bacteria tRNA *loci* are very uniform in terms of length (mean = 77.60 bp), when compared to archaea (mean = 83.10 bp) and eukaryotes (mean = 84.70 bp), which exhibit tRNA genes that are significantly longer (Figure 3J). Mature tRNAs, on the other hand, seem to follow an opposite pattern: bacterial mature tRNAs are significantly longer (median = 77.60 nt, IQR = 2.40) than what was observed in archaea (median = 76.80 nt, IQR = 1.92), with eukaryotes exhibiting significantly the shortest mature tRNAs with a very tight length variation (median = 74.50 nt, IQR = 0.67) (Figure 3J). The explanation for this opposite pattern possibly lies in the excessive number of tRNA genes with intronic sequences that are processed and removed from the mature tRNA in both archaea (18.99% of all tRNA *loci*) and eukarya (22.30%). tRNA genes containing any intronic sequences are exceedingly rare in bacteria, accounting for only ~0.11% (Figure 3K). In all domains, tRNA pseudogenes tend to be longer than functional tRNA genes. Interestingly, tRNA pseudogenes from bacteria are usually longer than those from archaea and eukaryotes (Figure 3J).

### 3.5. tRNA Intraspecific Copy Number Variation

After demonstrating that tRNA gene copy number is highly variable between domains and species, we turned to the amount of variation that can be observed within a species. Our data set contained multiple complete genomes for several species, which allowed us to further investigate the extent of variation that can be found between genes from the same species. We selected the top eight bacterial species with the greatest number of complete genomes, five eukaryotes (mostly fungi) and two archaea. Our analysis shows that all selected species in all domains exhibit intraspecific tRNA CNV (Figure 4A). For some species, the isolates varied more than two-fold in the number of tRNA *loci* (e.g., the yeast *Saccharomyces cerevisiae* and the bacterium *Salmonella enterica*). More importantly, tRNA intraspecific CNV does affect the number of available tRNA anticodons in the genomes of all domains (Figure 4B), indicating that this phenomenon might affect mRNA decoding.

### 3.6. tRNA-Derived Small RNA Fragments

Despite their central role in mRNA decoding into proteins, tRNAs are also a source of small RNA fragments (tRFs) that exhibit regulatory functions [41,42,43,44]. With that in mind, we asked how the observed range of variation in tRNA gene CN might influence the tRF pool that a cell is able to generate. We obtained tRF data from the tRFdb, a relational database of transfer-RNA-related fragments [35], which is based on the experimental identification of small RNAs in high-throughput sequencing data. The data set contained one bacterium (*Rhodobacter sphaeroides*) and seven eukaryotes, these being one fission yeast (*Schizosaccharomyces pombe*), two invertebrates, the roundworm (*Caenorhabditis elegans*) and the fruit fly (*Drosophila melanogaster*), and four vertebrates, namely, zebrafish (*Danio rerio*), African clawed frog (*Xenopus tropicalis*), mouse (*Mus musculus*) and human (*Homo sapiens*).

The total number of tRFs differs from one species to another and shows a strong positive correlation with the total number of tRNA *loci* per species (Figure 5A). If we consider only the number of unique tRF sequences per genome, the correlation with the tRNA gene count is cancelled (Figure 5B).

tRFs can be classified based on their original location on the mature tRNA [35,41]. We further explored the data with the tRFs classified into three categories: trailer-derived tRF (tRF-1), 3′-derived tRF (tRF-3) and 5′-derived tRF (tRF-5) (Figure 5). tRF length variation ranged between 13 to 38 nt (Figure 5C). Non-redundant tRF length distribution varied considerably between species (Figure 5C). Distinctive patterns were observed even between vertebrates, such as the African clawed frog, with most tRFs ranging between 15 and 22 nt, while the majority of mouse tRFs ranged between 18 and 22 nt (Figure 5C). Notably, the fission yeast produces the longest fragments, a pool mostly composed of tRF-3 (Figure 5C). Interestingly, the mouse and human (both mammals) show a very similar distribution of tRF size, with a peak at 18 and 22 nt.

The total number of non-redundant tRFs per species varies considerably. Surprisingly, the data suggest that the organism with the less diverse pool of tRFs in our sample is the roundworm (47 unique tRFs), with ~19% fewer non-redundant tRFs than the bacterium *R. sphaeroides* (58 unique tRFs) (Figure 5D). Between vertebrates, the African clawed frog has a strikingly low number of unique tRFs (51 non-redundant tRFs), which represents about 40% of the tRF pool found in the fruit fly and zebrafish (86 and 87 non-redundant tRFs, respectively) and only 1/3 of the mouse non-redundant pool (159 tRFs) (Figure 5D). Interestingly, African clawed frog and zebrafish are among the vertebrates with the highest CN of tRNA, with 3457 and 10,471 tRNA *loci*, respectively. This indicates that the pool of tRFs these organisms produce is highly redundant in terms of sequence and size (Figure 5E). Over 98% of the tRFs identified in zebrafish are redundant. At the other extreme, the bacterium *R. sphaeroides* tRFs have a very low sequence redundancy, with nearly 70% of the identified tRFs having a unique sequence (Figure 5E).

Finally, we quantified the amount of unique tRFs per type, showing that the distribution of unique fragments varies considerably depending on the genome (Figure 5F). The mammals (human and mouse) seem to produce more unique tRF-3 and tRF-5 fragments, in this order, a trait that is conserved in all the other species evaluated (Figure 5F). While the African clawed frog has more tRF-5 and tRF-1 fragments, the zebrafish is the only species that produces more unique trailer-derived tRF (tRF-1) fragments. The roundworm seems not to produce any tRF-1 fragments (Figure 5F). This lack of tRF-1 small RNAs is not observed in any of the other eukaryotes evaluated, including the unicellular fission yeast (Figure 5F). Interestingly, we did not detect tRF-1 fragments in the bacterium *R. sphaeroides* (Figure 5F). Thus, the production and accumulation of 5′- and 3′-derived tRFs seems to be conserved between different domains, while trailer-derived tRFs seems to be absent in at least some species of bacteria and eukarya.

## 4. Discussion

Our results confirm previous findings that eukaryote genomes have substantially more tRNA genes than prokaryotes [1,2,16,17,18,20,21,32,33,45]. In addition, we found that eukaryotes show a tRNA CNV of three orders of magnitude between species, compared to one and a half in bacteria and less than one in archaea (Figure 1). This variation is well correlated with genome size. In our archaea sample, variation in genome size was from ~1 to ~6 Mb, or half an order of magnitude. Bacterial genomes varied by one and a half orders of magnitude, from ~0.5 to ~10 Mb. Eukaryotic genomes varied by up to four orders of magnitude, from ~10 Mb to ~100 Gb. These observations indicate that genome size fluctuations are a major player in shaping the number of tRNA genes. Eukaryotes are the only known organisms to evolve genomes >>10 Mb, with some vertebrates and land plants reaching genomes >>1 Gb, such as salamander species with ~120 Gb (1C = ~120 pg) [46], lungfishes with ~130 Gb (1C = ~133 pg) [47] and whisk-ferns and some monocots up to 150 Gb (1C = ~151 and ~152 pg, respectively) [48]. tRNA gene count seems to have evolved along with the genome size expansion that occurred in eukaryotes that is mostly explained by the expansion of non-coding DNA (Figure 3).

Genomes containing large sets of tRNAs (>>500 *loci*) belong to taxa that emerged from the Neoproterozoic onwards (last billion years), with the early diversification of animals and green plants [49,50]. Eukaryotes with high cellular diversity (i.e., land plants and vertebrates) also have a more diverse pool of tRNA isoacceptors (between 50 and 60), when compared to bacteria, which tend to encode a tRNA pool with 30 to 40 different anticodons. Besides generally having fewer tRNA genes packed in smaller genomes, archaea have a diversity of tRNA anticodons that is comparable to microbial eukaryotes with small cell diversity (i.e., fungi and ‘protists’), between 40 and 50 different anticodons (Figure 3). This indicates that the first eukaryote common ancestor (FECA) likely had a more diverse set of tRNAs than typical extant bacteria. Indeed, the association of anticodons and amino acids are more similar between eukaryotes and archaea (Figure 2). These findings suggest that after the early split between bacteria and archaea/eukarya, a broader set of anticodons evolved in the archaeal lineage even before the evolution of eukaryotes. Later, with the independent evolution of macroscopic eukaryotic lineages with a high diversity of cell types (i.e., plants and animals), the pool of tRNA anticodons tended to approach the combinatorial limit of DNA triples around 60 anticodons (excluding one to three stops), in opposition to generally 30 to 40 in bacteria. The lower diversity of tRNA anticodons in prokaryotes and unicellular eukaryotes when compared to complex eukaryotes indicates that they might rely more on wobble decoding since they presumably use most of the 61 coding codons in their protein coding mRNAs. It is possible that the ribosome of bacteria is more permissive for wobble pairing than the ribosome of complex eukaryotes, which could help explain why plants and animals independently evolved a high diversity of tRNA anticodons that could lead to more precise pairing of mRNA codons to tRNA anticodons. Interestingly, archaea and single-celled eukaryotes seem to demand a higher number of tRNA anticodons than most bacteria but might still make use of wobble pairing more frequently than complex macroscopic eukaryotes such as plants and animals.

Our results seem to indicate that complex macroscopic organisms require a vaster anticodon diversity coupled with high tRNA gene redundancy provided by high CN. This result may help explain why highly complex multicellular life emerged only in the archaeal lineage, even with the astonishing diversity of bacteria. Besides important differences in tRNA anticodon diversity, the CN rank of tRNA genes dedicated to transporting each amino acid is mostly conserved among the three domains (Appendix A), suggesting that some amino acids might demand more tRNA gene copies in all kinds of cells. The intrinsic features of the tRNAs, such as short length, base modifications, the various protein binding partners and tRNA fragmentation events, make tRNA a very versatile molecule found in multiple cellular processes beyond the protein synthesis [51]. These multi-tasking features of tRNAs arose from the basal role of tRNAs and might have played an important role in the diversification of the tRNA repertoire in the early evolution of cellular life.

While eukaryotic and prokaryotic genomes are clearly distinguishable in terms of protein-coding gene density per base, fungi species have genomes with a tRNA gene density comparable to prokaryotes. At the other extreme, land plant and vertebrate genomes have very low tRNA gene density. Even eukaryotes with extreme tRNA CN, such as the zebrafish (10,471 tRNA genes), have ~7 tRNA genes/Mb, which is comparable to the median observed for fungal genomes. Our results indicate that, besides having up to three orders of magnitude fewer tRNA genes when compared to eukaryotes, prokaryote genomes are much denser in tRNA *loci* per Mb.

Organisms vary in terms of the combinations of anticodons and amino acids that are present in the tRNA pool. In our sample, we observed that 63 out of the 64 possible anticodons are associated with the transport of amino acids in at least some genomes. Interestingly, only the tRNA genes bearing the anticodon TTA were never found associated with any amino acid in any genome. TTA anticodons correspond to the ochre stop codon (UAA) [52], and although we did not find any species that uses this stop codon to transport any amino acid in our sample, Kuchino et al. 1985 reported tRNA(UAA) transporting the amino acid glutamine in the ciliate *Tetrahymena thermophila* [53]. However, the few species related to this ciliate in our sample did not show this feature, suggesting that this association is likely rare. Our results suggest that the anticodon TTA might be the ancestral stop codon since it is the only codon that is avoided in the transport of amino acids in virtually all organisms. We found that 19 amino acids out of the standard 20 are associated with preferential anticodons that are very pervasive in all domains (Figure 2). Isoleucine is the only exception among standard amino acids, mostly due to the evolution of *Eukarya*-specific anticodon. In our sample, tRNAs that transport selenocysteine are rare in all domains. The general positive correlation between the CN of tRNAs dedicated to transport each amino acid in all domains is cancelled when considering the CN of tRNAs classified by anticodon (Appendix A), with ^Met^tRNA_(CAT)_ being the only one exception. ^Met^tRNA_(CAT)_ genes are present in high CN in all domains, suggesting that there is a specific universal advantage in gene redundancy when it comes to the tRNA that usually starts protein decoding, a phenomenon that is not observed for any other tRNA gene. The reason must be further explored; however, it is tempting to speculate that more gene copies could be needed to achieve a high cellular concentration of ^Met^tRNA_(CAT)_ which might be important for protein synthesis. Our data suggest that eukaryotes and archaea tRNAome are more similar in terms of the combinations of anticodons and amino acids, most likely due to their common ancestry (Figure 2).

Our results indicate that the tRNA gene pool evolved concerted with genome size in a broad macroevolutionary pattern unlike what is observed for protein-coding genes (Figure 3). Overall, tRNA CN increases as genome size increases, while protein-coding genes seem to have an upper limit of ~35,000 genes per haploid genome, independent of genome size. In fact, the number of protein-coding genes is tightly correlated with genome size in prokaryotes. In eukaryotes, genome gigantism (>>100 Mb) is not coupled with an equivalent accumulation of protein-coding genes. This leads us to conclude that the accumulation of tRNAs genes and protein-coding genes in genomes follow different macroevolutionary patterns. Similarly to functional tRNAs, tRNA pseudogene count per genome also increases with genome size in a similar fashion (Figure 3). This is likely explained by the fact that genomes with high counts of tRNA genes might accumulate more tRNA pseudogenes over time. Gene structure, however, is different. tRNA pseudogenes are longer in all domains, in line with tRNAs facing selective pressure against the insertion of additional nucleotides cancelled in pseudogenes.

tRNA gene analysis indicates that eukaryotes tend to have a longer genomic *locus* in comparison to bacteria, while archaea genes have an intermediate length. The difference in length is explained by the higher prevalence of introns in tRNA genes in archaea and eukarya when compared to bacteria, where tRNA introns are exceedingly rare. By contrast, mature tRNAs of bacteria are longer than their eukaryotic counterparts, again with archaea having an intermediate length. These findings indicate that the evolution of archaea and the eukaryogenesis itself might have involved a slight average increase in tRNA intron length by 7 to 8 nt per gene, coupled with the elimination of an average of 4 nt in mature tRNAs when compared to bacterial counterparts. This fine tuning of tRNA might be related to an increased protein translation efficiency in eukaryotes [54].

Considering the role of tRNAs in translation, CNV in the tRNA gene pool may affect translation efficiency [2,10,18,19,20,21]. With that in mind, we selected species in our dataset for which a significant number of genomes have been sequenced to check whether intraspecific CNV of tRNAs are frequently observed. All species from the three domains showed evidence of tRNA CNV between individuals. In all cases tRNA CNV does affect the anticodon repertoire, which might directly impact codon usage and translation efficiency [20,28,55]. These results suggest that tRNA CNV is likely a widespread phenomenon that also takes place at a microevolutionary scale, accounting for differences between strains and likely between individuals. Further investigation is needed to assess the degree of variation between individuals of eukaryotes with large genomes as land plants and vertebrates. However, our data using the complete genome of 17 strains and subspecies of mice (*M. musculus*) indicate that mammals display intraspecific variation in tRNA gene CN that can even affect the anticodon repertoire, as seen for the wild-derived strain PWK/PhJ (*M. musculus musculus*), which has one fewer isoacceptor compared to the other 16 strains (Figure 4).

Apart from their central role in translation, tRNAs are also subject to enzymatic cleavages that lead to the accumulation of tRNA-derived RNA fragments (tRFs) [35], a class of small RNAs with regulatory functions. We gathered evidence from eight species showing that variation in the tRNAome between lineages is also coupled with pronounced variation in the pool of tRFs, which might have a regulatory impact. tRFs are produced by eukaryotes and prokaryotes, suggesting that the processing of tRNA into small RNAs is a widespread phenomenon. Considering the pool of unique tRFs found in each species, we observed that no strong relationship could be found between tRNA gene CN and the diversity of tRFs. In our sample, the mammals were able to produce more than 130 unique tRFs, while we only found 47 in *C. elegans*. A high degree of sequence redundancy between tRFs was observed for the eukaryotes, especially those with high tRNA CN, such as the zebrafish and the African clawed frog. In bacteria, it seems that most tRFs have non-redundant sequences, possibly due to the low tRNA gene redundancy. Our data indicate that some species such as the bacterium *Rhodobacter* and the roundworm *C. elegans* lack the ability to process and accumulate tRFs that are derived from the trailer sequence present in the primary tRNA transcript. It is possible that *C. elegans* lost the ability to produce tRF-1 since they are found in fission yeast and the other animals. Moreover, we observed a variation in the distribution of tRF size between species, with most fragments within a range of 14 to 34 nt. tRFs derived from the 5′ of the mature tRNA tend to be longer, with a median size of ~22 nt, while tRF-1 and tRF-3 have an overall similar size distribution with a median length around 19 nt (Appendix A). Interestingly, eukaryotes that produce longer tRFs, such as the fission yeast (mean length = ~24 nt), tend to have tRF sequences with a more balanced composition of bases, while species with shorter tRFs tend to produce fragments with a high CG composition (Appendix A), indicating a possible sequence bias between species. Humans and mice share an interesting common pattern of tRF diversity. Most of their tRFs seem to fall into two classes of size, tRF-3 with 18 and 22 nt (Figure 5C). More detailed studies are needed to understand the regulatory outcomes of such variability in the tRF pool of cells.

Our study revealed macroevolutionary patterns in the evolution of tRNA genes. We showed that tRNA gene expansion requires genome size increase, which possibly explains why prokaryote genomes of all lineages are usually unable to bear >100 tRNA genes, while genomes of animals and plants can bear up to tens of thousands. The increased genome size and protein-coding gene pool of eukaryotes evolved coupled with an increase in tRNA gene length and CN, a decrease in mature tRNA length and an expansion in the repertoire of tRNA anticodons. Taken together, our findings suggest that the evolution of tRNA genes played important roles in the early split between bacteria and archaea and in the later origin and diversification of eukaryotes, from simple archaeal ancestors to highly complex animals and plants.

## Figures and Tables

**Figure 1 genes-14-00027-f001:**
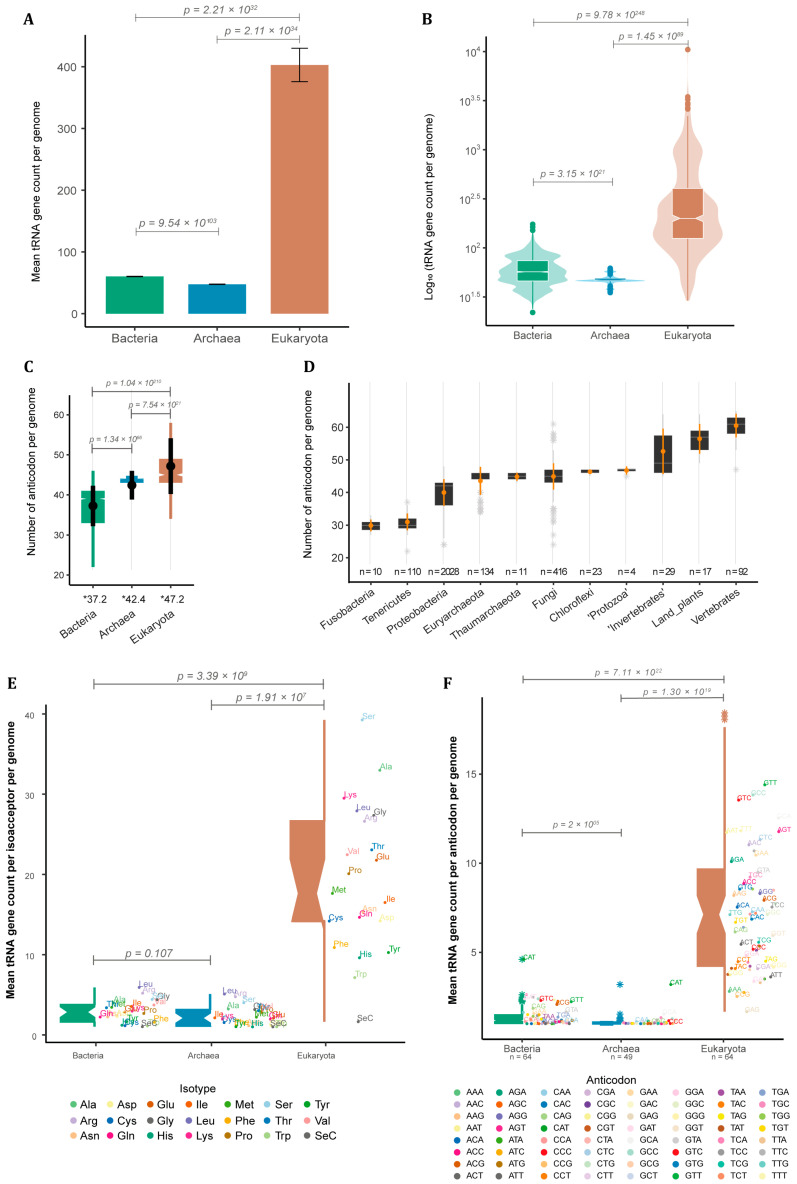
tRNA gene count across the three domains of life. (**A**) Mean tRNA gene count per genome with standard error (SE) and (**B**) distribution of tRNA gene count per genome in bacteria (*n* = 4027), archaea (*n* = 217) and eukarya (*n* = 559). (**C**) Distribution of the number of tRNA isoacceptors (different anticodons) per genome per domain, with mean and standard deviation (SD) in black (* = mean of anticodon per genome per domain). (**D**) Distribution of tRNA isoacceptor count of selected taxa, with mean and SD in orange (*n* = number of genomes per taxa). (**E**) Number of tRNA genes per amino acid and (**F**) anticodon per genome per domain (*n* = number of different anticodons). *p*-values are shown for all contrasts (see methods for statistical analyses).

**Figure 2 genes-14-00027-f002:**
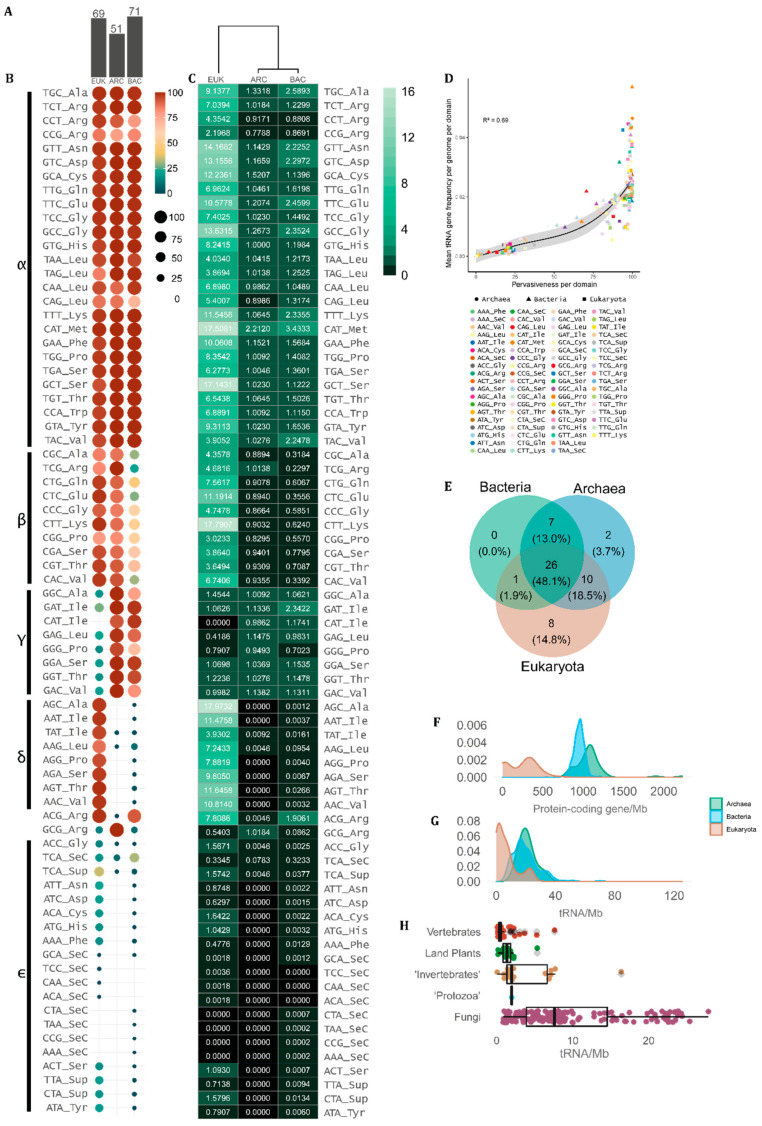
Occurrence and frequency tRNA amino acid/anticodon combinations in the genome of cellular organisms. (**A**) Number of tRNA combinations observed between amino acids and anticodons per domain (EUK = eukarya, ARC = archaea, BAC = bacteria). (**B**) tRNA pervasiveness (% of genomes) and (**C**) mean tRNA count per domain. (**D**) Correlation between tRNA mean gene frequency per genome per domain and tRNA pervasiveness defined as the proportion of genomes containing each tRNA per domain. (**E**) Venn diagram showing the overlap between domains of the tRNA anticodons present in more than 70% of the genomes in each domain. (**F**) Distribution of the number of protein-coding and (**G**) tRNA genes per megabase (Mb) per genome per domain. (**H**) Boxplots showing the distribution of tRNA genomic density (tRNA genes/Mb) per genome in the main eukaryote groups.

**Figure 3 genes-14-00027-f003:**
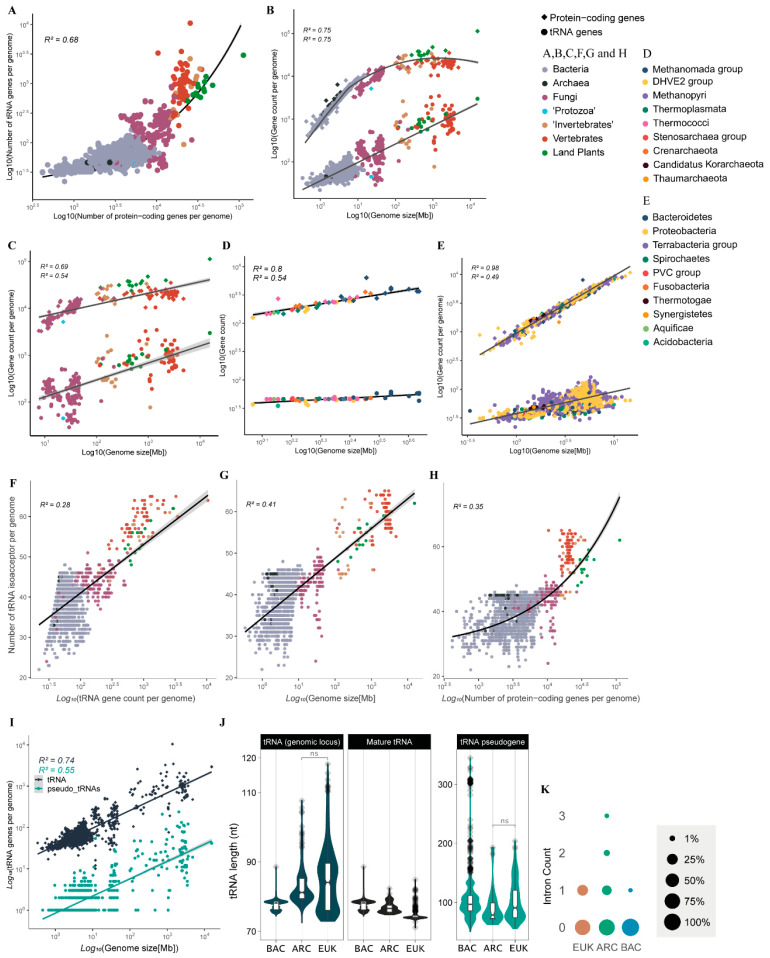
Macroevolutionary patterns of tRNA copy number and diversity across cellular organisms. (**A**) Correlation between the number of tRNA genes and protein-coding genes per genome. (**B**) Correlations between genome size and the number of tRNA genes and protein-coding genes per genome. The same as (**B**) for (**C**) eukarya, (**D**) archaea and (**E**) bacteria. Correlation between count of tRNA isoacceptors and number of tRNA genes per genome (**F**), genome size (**G**), and number of protein-coding genes per genome (**H**). (**I**) Correlation between the number of tRNA pseudogenes (along with tRNA functional genes) and genome size. (**J**) Distribution of length (nt) of tRNA genomic *loci*, mature tRNA and tRNA pseudogenes per domain, showing statistically significant differences, except for the non-significative contrasts between archaea and eukarya (ns, *p*-value > 0.05; see methods for statistical details). (**K**) Frequency of tRNA genes containing a given number of introns per domain.

**Figure 4 genes-14-00027-f004:**
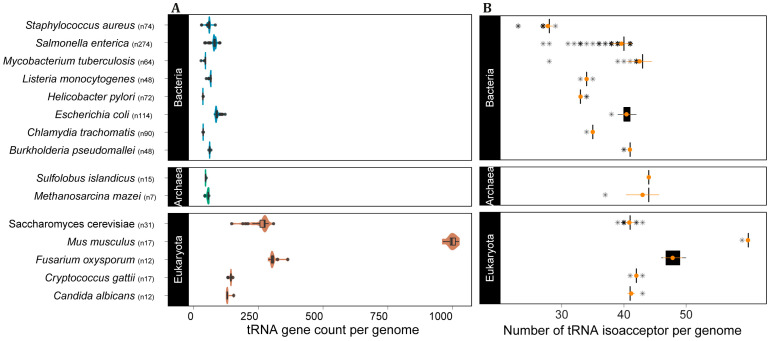
Intraspecific tRNA CNV affects tRNA anticodon repertoire in the three domains of cellular life. (**A**) Distribution of tRNA gene count per genome per species and (**B**) number of tRNA anticodons per genome per species, with mean and SD in orange, the asterisks represent the boxplot outliers. *n* = number of genomes per species.

**Figure 5 genes-14-00027-f005:**
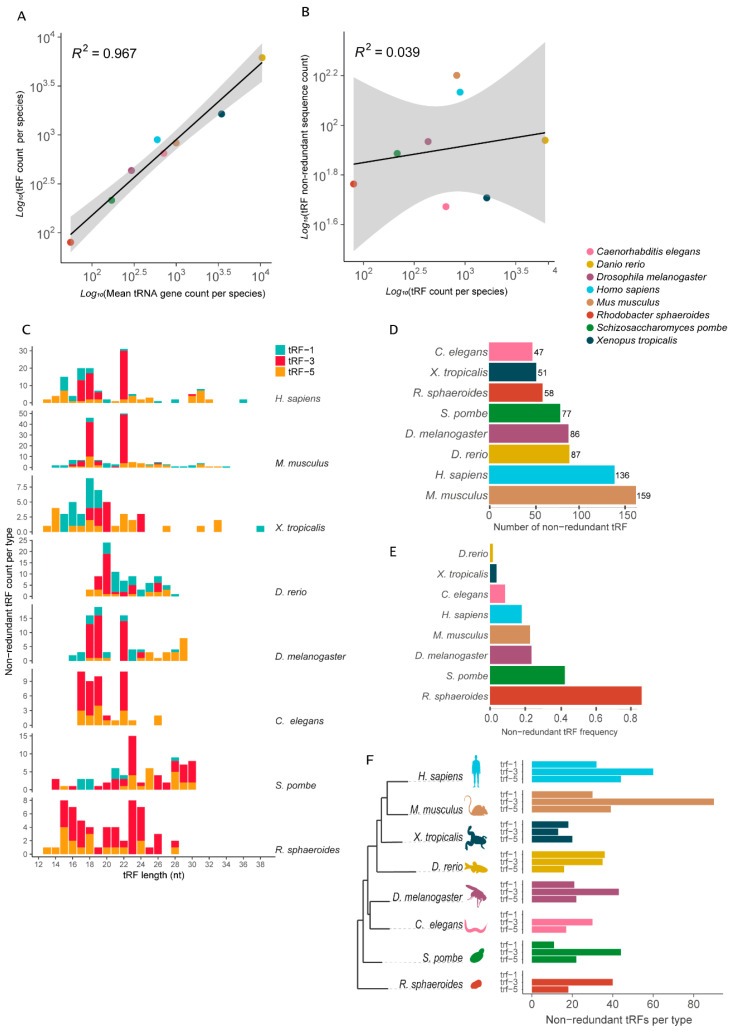
Diversity of tRNA-derived RNA fragments (tRFs) attributes in selected genomes. (**A**) Positive correlation between tRF count and tRNA gene count per species. (**B**) Absence of correlation between non-redundant tRF count and total tRF count between species. (**C**) Distribution of tRF length per type per species (nt). (**D**) Total number of non-redundant tRFs per species. (**E**) Frequency of non-redundant tRFs (non-redundant tRFs/total number of tRFs) per species. (**F**) Number of non-redundant tRFs per type per species in a phylogenetic perspective.

## Data Availability

All scripts with detailed usage instructions and intermediate files are available at github.com/fenibrito/tRNA_cnv accessed on 14 December 2022.

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
