# Peer review of "The Evolution of tRNA Copy Number and Repertoire in Cellular Life"

_genes, 2022, doi:10.3390/genes14010027_

Round 1
Reviewer 1 Report
Genes, 2022, #2070981
Santos and Del-Bem’s pan-genome tRNA analysis is graphically pleasing and an overall well written study, that is very descriptive but an appropriate fit for Genes and a good resource for the community.
Major points:
· The tRF analysis is timely and neat. However, as the authors themselves point out, tRFs come in different classes, similar to other regulatory, small RNAs (e.g. miRNA versus piRNA etc.). For example, tRF5 and tRF3 comprise distinct small RNA classes with distinct functions and sizes (tRF5a, tRF5b, … tRF3b, tRF3a, etc. see Kumar et al. 2016, PMID: 27263052), visible in Figure 5F. Therefore, ‘mean/median tRF length per species’ (Figures 5 B-D) is not really biological meaningful and should be avoided. Figures 5 B-D could be split up into these different classes in each species or show length in bins or sliding windows (e.g. 16-19 nt, 20-23 nt, …).
Minor points:
· Figures 3 B-E: round versus diamond shaped dots are impossible to tell apart. The two regression lines should be labeled or dot size increased. (This will make it easier to see whether ‘tRNAs CN expands faster than protein-coding genes’.)
· Line 156: How would the authors know about unusual anticodons transporting amino acids other than predicted? Similar line 398: "pervasive associations between anticodons and amino acids" What is meant by association (experimental?)? Perhaps: The number of anticodons per amino acid are more similar between...
· When discussing sequence redundancy (lines 349-361) the authors should consider whether perhaps small RNA data sets in the database were limited for some species? It would be also worth mentioning that a confounding factor for redundancy or library depth might be how many other small RNAs are expressed in that organism and tissue. For example, bacteria have no miRNA, so sequencing small RNA will result in much more depth for tRF, while e.g. many zebrafish tissues express abundant other small RNAs (miRNA, piRNA) that effectively compress the sequence space or uniqueness of tRFs.
· Please correct lines 424-426: tRNAs with the rare TTA anticodon have been found in Actinobacteria and Tetrahymena (PMID: 21773800, PMID: 2991890) and perhaps other organisms.
· The authors could elaborate a bit on why higher organisms have a larger repertoire of anticodons? Schimmel 2017 (PMID: 28875994) discusses a number of explanations, one of them that "gene-to-gene variation in codon usage offers a way to modulate relative translation efficiencies among various mRNAs".
· It is too hard to tell apart anticodons in Supplemental Figure 1B (colors too similar). Label isoacceptors (same aa) with same color plus text labels?
· Lines 454-455: It is somewhat trivial that tRNA genes are under selective pressure compared to pseudogenes. Maybe replace "suggesting that" with "in line with tRNAs facing selective pressure...."?
· Line 474: CNV between individuals is very speculative... but could be tested by analyzing 1000Genomes data.
· Grammar or typos:
line 64 different not “diverse”?
line 235 “increases”?
line 138 in contrast not “in opposition”
line 304 “indicating that this phenomenon might affect mRNA decoding into proteins” is unclear
line 349 “61”?
line 381 "up to two orders"?
Reviewer 2 Report
This manuscript described the evolution of tRNA copy and repertoire in diverse cells. The author analyzed the tRNAs from thousands of species and indicated that the tRNA gene pool experienced increased anticodon diversity after Bacteria/Archaea split. They also demonstrated that the evolution of tRNA genes played important roles in macroevolutionary. Otherwise, all these finds are highly supported by the tRNA database and annotations. These results are interesting for tRNA evolution. However, there are some limitations in the present manuscript.
Concerns:
- For Results 3.1 and Figure 1A, the manuscript mentioned that eukaryotes n=525, however, there are n=559 in Figure 1A. Also, in manuscript lines 95-96, “eukaryotes have a mean of 403 ± 27.1 tRNA loci per genome …” which there are no more than 400 in Figure 1A, same as the Bacteria part, the figure’s data about the SD is not consistent with the manuscript. Besides, how can bacteria with a mean = 60.30 ± 20.00 SD VS Archaea with a mean = 47.60 ± 5.41 SD have such a significant difference?
- There is inappropriate to have T in the RNA part, RNA has no T but U. Also, if for codon in DNA sequence, it can be TTA, otherwise, tRNA-UUA might be better.
- Line 240-242. “For eukaryotes, our data suggest that tRNAs CN expands faster than protein-coding genes when the genome increases in size, mostly due to whole genome duplications (WGD) (Figure 3C).” It might be due to the non-coding part of the genome in eukaryotic cells. It would be better to include the analysis data about the count of tRNAs per Coding DNA Sequence size for each species despite the whole size of the genome. It may differ and might give a different insight into the tRNA count numbers and evolutions.
- 286-288 “intronic sequences that are processed and removed from the mature tRNA in both, Archaea (18.99% of all tRNA loci) and Eukarya (22.30%)”. What’s the difference between the intronic sequence and tRFs? tRFs are rarely found in Bacteria consistent with the data about the intronic size. The manuscript should include the tRFs of Archaea and compare them to the others. And lastly, the patterns of the intronic sequence evolution part should be discussed with more publications.
Reviewer 3 Report
The authors present a comprehensive analysis of the tRNA copy number landscape based on currently known sequenced and annotated genomes. Using data contained within well-curated databases, the authors show correllations between tRNA loci/diversity and different domains/phyla. The results presented are of high quality, but would benefit from improvements in presentation and description. My concerns/suggestions follow:
1. In the Materials and Methods, for differences to be considered significant, the p-values should be < 0.05, not > 0.05.
2. There appear to be inconsistencies between the numbers in the text of page 3 and those depicted in Figure 1A. For Eukaryotes, the average should be 403, but the bar only appears to be somewhere in the vicinity of 360. Also, the error bars for Bacteria and Archaea should be 20 and 5.41, but do not show as that large in the figure. Please revise accordingly.
3. Many of the p-values in Figure 1 appear absurdely low. Considering the visible overlap in data points for Bacteria and Archaea, the p-values would be extected be significantly larger than 10^-102. Please check that these values are correct.
4. It is difficult to tell the difference between the points in Figures 1E and 1F. It would be helpful if some of the less bunched-up data points were labeled.
5. A lower number of anticodons represented in certain organisms may be indicative of important biological factors. For example, these organisms presumably still use all or nearly all of the 61 coding codons; therefore, something biological may play a role in allowing for expansion of decoding ability in tRNAs. tRNA modifications or more flexibility in allowing wobble decoding in these ribosomes are attractive mechanisms.
6. Does the analysis presented include mitochondrial tRNAs? If so, it may be helpful to differentiate these from their cytoplasmic counterparts. If they were not included, it would be nice to see how these compare.
7. How were the tRNA pseudogenes identified in this study?
8. Overall, some of the results were difficult to follow and required several re-readings to understand. It would have been helpful if terms such as "anticodons per genome" and "tRNA anticodon diversity". These can be read a bit ambiguously. It was unclear while reading if anticodons per genome included modified nucleotides or even the number of times that particular anticodon triplet was found in the genome sequences. Similarly, it took some digging through the figures and text to understand precisely what tRNA anticodon diversity was referring to. Specifically defining these terms would have made the text eminently more readable.
Round 2
Reviewer 2 Report
The authors have addressed my concerns.